# Root Ethylene and Abscisic Acid Responses to Flooding Stress in *Styrax japonicus*: A Transcriptomic Perspective

**DOI:** 10.3390/plants14121870

**Published:** 2025-06-18

**Authors:** Chao Han, Jinghan Dong, Gaoyuan Zhang, Qinglin Zhu, Fangyuan Yu

**Affiliations:** Collaborative Innovation Centre of Sustainable Forestry in Southern China, College of Forest Science, Nanjing Forestry University (NJFU), Nanjing 210037, China; hanc0909@njfu.edu.cn (C.H.); djh01@njfu.edu.cn (J.D.); zhanggaoyuan@njfu.edu.cn (G.Z.); 8220110118@njfu.edu.cn (Q.Z.)

**Keywords:** root, phytohormone, pathway, flooding, *Styrax*

## Abstract

Global climate change has led to an increased frequency of extreme weather events, with flooding caused by heavy rainfall posing a significant threat to plant growth and survival. *Styrax japonicus*, a species of ecological and economic importance, exhibits stronger flooding tolerance compared to its congener *Styrax tonkinensis*. Endogenous hormonal systems in plants are indispensable for integrating growth dynamics, developmental transitions, and ecological stress perception-transduction pathways. To investigate the response of *S. japonicus* to flooding stress at both hormonal and molecular levels, this study utilized annual seedlings of *S. japonicus* as experimental material. Two levels of flooding stress, waterlogging and submergence, were applied to examine the variations in endogenous hormone levels in *S. japonicus* roots under different stress conditions and durations. Combined with transcriptome sequencing, critical genes associated with hormone-mediated signaling and biosynthetic processes were identified. The results showed that the content of the ethylene precursor ACC exhibited a trend of “increase–decrease–increase”, with an earlier decline under submergence compared to waterlogging stress by approximately 10 days. Abscisic acid content sharply decreased at 5 d, followed by an initial increase and subsequent decrease, with higher ABA levels observed under waterlogging stress than under submergence. GA content significantly decreased after 10 d in both stress conditions. KEGG enrichment analysis revealed that the most prominently enriched pathway for DEGs was plant hormone signal transduction under both waterlogging and submergence stress, with 314 and 370 DEGs identified, respectively. Analysis of common genes indicated their association with ethylene, ABA, auxin, and BRs. After further investigation of DEGs in the ethylene and ABA biosynthesis process, we identified key enzyme genes encoding ACS, ACO, and NCED, which are critical for their biosynthesis.

## 1. Introduction

*Styrax japonicus*, a shrub or small tree of the genus *Styrax* in the family Styracaceae, is the most widely distributed species of its genus in China [1]. As a heliophilic species, it exhibits rapid growth [2]. Studies have demonstrated that various parts of *S. japonicus*, including its leaves, fruits, flowers, resin and roots possess significant medicinal properties [3,4]. Its flowers are mostly white and have a light fragrance, making it an excellent ornamental tree [5]. Furthermore, *S. japonicus* has multifunctional values, encompassing essential oil production, timber utilization, and ecological conservation benefits [6,7]. However, current research on *S. japonicus* and other species in the *Styrax* genus remains limited. This scarcity of scientific investigation has hindered the widespread cultivation and practical application of *Styrax* species, while also posing challenges to the conservation of their germplasm resources.

In addition, as the climate has been warming, extreme and heavy precipitation frequently strikes across the globe [8]. Based on meteorological data collected from national observation stations operated by the China Meteorological Administration’s National Meteorological Center, statistical analysis reveals that between 2012 and 15 September 2023, a total of 503 torrential rainfall events were recorded across China. Notably, the southern region of China has emerged as the area with the highest frequency of such extreme precipitation events during this period (https://www.cma.gov.cn, accessed on 6 March 2025). *S. japonicus* is predominantly distributed in southern China, where its growth and development are significantly threatened by flooding stress caused by heavy precipitation events.

During the research conducted by our team, it was observed that, in the Nanjing region, *S. japonicus* exhibits stronger growth vigor compared to its congener, *Styrax tonkinensis*. Specifically, when exposed to water-saturated environments (2–5 cm above the substrate surface) for 72 h, the leaves of one-year-old *S. tonkinensis* seedlings showed signs of wilting and drooping [9]. Nevertheless, one-year-old *S. japonicus* seedlings did not exhibit similar symptoms until 15–20 days of waterlogging (4 cm above the substrate surface). To better understand how *S. japonicus* responds to flooding stress, we seek to investigate the possible reasons for its superior tolerance compared to *S. tonkinensis*.

Plant hormones are signaling compounds that serve as key regulators in plant responses to environmental stress, abscisic acid (ABA), auxin, brassinosteroid (BR), cytokinin (CK), ethylene (ET), gibberellic acid (GA), jasmonic acid (JA), salicylic acid (SA) and strigolactones (SL) are widely recognized as key phytohormones that play crucial roles in regulating plant responses to environmental stimuli [10]. The rapid accumulation of ethylene serves as the primary signal triggering adaptive responses in plants to flooding, which involves the transformation of S-adenosylmethionine (SAM) into 1-aminocyclopropane-1-carboxylic acid (ACC), facilitated by ACC synthase (ACS), and followed by the oxidation of ACC to ethylene through the action of ACC oxidase (ACO) in the presence of oxygen [11,12]. The ethylene molecule, upon binding to its receptor on the endoplasmic reticulum membrane, typically inhibits the activity of CTR1, and then it allows downstream transcription factors, such as EIN2 and EIN3, to become activated, thereby regulating the activation of ethylene related gene expression [13]. Experimental evidence indicates that waterlogging induces ethylene accumulation across various angiosperm species, and it enhances survival rates during flooding, hypoxia, and reoxygenation stress through mechanisms such as signal transduction, mediation of metabolic processes, mitigation of reactive oxygen species damage, and induction of autophagy [12].

Abscisic acid (ABA), acting downstream of ethylene, modulates the plant’s response to hypoxic stress [14]. It participates in the development of hyponastic growth during hypoxia, adventitious roots, and the creation of secondary aerenchyma [15]. Research indicates that certain plant species, such as tomatoes and citrus, accelerate growth to evade submergence stress through a rapid decline in endogenous ABA levels, while hypoxia-sensitive plants like *Pisum sativum* and *Lactuca sativa* typically respond to oxygen deprivation by significantly elevating ABA concentrations [16,17,18,19,20]. Studies have shown that ABA and gibberellin (GA) exert opposing influences on various aspects plant development [21]. In rice, ethylene decreases the level of ABA, amplifies tissue responsiveness to GA and encourages the growth of submerged internodal regions [22]. Similarly, it was found that elevated ethylene concentrations inhibited endogenous levels of ABA but stimulated GA biosynthesis to induce cell elongation [23].

Therefore, this study selected current-year seedlings of *S. japonicus* and set two flooding depths to simulate waterlogging and partial submergence, respectively. Root samples were collected on 0 d (as the control) and 5 d, 10 d, 15 d, 20 d, 25 d after treatment (Figure 1). Then they were used to measure endogenous hormone levels and perform transcriptome sequencing. Subsequently, transcriptomic data combined with changes in plant endogenous hormone levels were analyzed to investigate the hormonal response of *S. japonicus* roots to flooding stress. Furthermore, several critical genes involved in plant hormone signal transduction and biosynthesis pathways were pinpointed.

## 2. Results

### 2.1. Changes in the Content of Endogenous Plant Hormones in S. japonicus Roots Under Flooding Stress

#### 2.1.1. ACC

Under prolonged flooding conditions (25 days), the content of ACC decreased in the middle and later stages of stress (Figure 2). Notably, under waterlogging stress, the ACC content in the roots showed a significant decline at 25 d; moreover, the values at 5 d, 10 d, 15 d and 20 d were higher than that at 0 d. Under submergence, the highest value was recorded at 10 d and subsequently remained lower than that at 0 d. It is noteworthy that the ACC content under submergence was consistently lower than that under waterlogging stress throughout the experimental period, and the decrease in ACC content was approximately 10 days earlier compared to waterlogging stress.

#### 2.1.2. ABA, GA1, SA and T-Zeatin

As illustrated in Figure 3a, the ABA content in the roots of *S. japonicus* seedlings exhibited a sharp decline following flooding stress treatment, manifesting a pattern characterized by an initial rise preceding a subsequent decline, with values significantly lower than those at 0 d. The ABA content under waterlogging is higher than that under submergence. Under waterlogging stress, the GA1 content in the roots reached its peak of 10.16 ng/g at 10 d, while at other time points, it remained below the levels observed at 0 d. In addition, the content of GA1 began to decline significantly at 10 d, as with under waterlogging stress (Figure 3b). The SA content in the roots under both treatments showed a similar pattern, with the lowest SA content observed at 5 d under waterlogging stress (1.533 ng/g), and at 10 d under submergence (1.98 ng/g) (Figure 3c). The T-zeatin content in the roots under flooding stress did not show a clear pattern, but it significantly decreased at 5 d and equalized at 15 d, with a value of 0.02 ng/g (Figure 3d).

### 2.2. De Novo Assembly and Quality Control

33 cDNA libraries from the roots of *S. japonicus* subjected to flooding stress at 0 d, 5 d, 10 d, 15 d, 20 d and 25 d were constructed for RNA-Seq sequencing utilizing the Illumina platform. We acquired a total of 1472.71 M high-quality reads and 218 G valid nucleotide data, with a Q30% of 95.88% and an average GC content of 46.66% (Appendix A). Following filtration, the Trinity software was employed for De novo transcriptome assembly to ensure comprehensive and accurate reconstruction of the transcript sequences.

### 2.3. PCA Analysis

To assess the overall variation, reproducibility among biological replicates and identify potential outliers, PCA was performed. Under conditions of waterlogging stress, PC1 and PC2 explained 60.18% and 14.27% of the total variance, respectively, while under submergence, PC1 and PC2 contributed 43.29% and 16.63%, respectively (Figure 4). A clear separation was observed between the treatment and control groups, with good reproducibility within the sample groups.

### 2.4. KEGG Analysis of DEGs

KEGG pathway analysis was conducted on DEGs identified from five experimental groups exposed to waterlogging and submergence conditions, with 0 d serving as the reference control. The analysis revealed that waterlogging stress primarily induced significant enrichment of DEGs in key pathways, including “Plant hormone signal transduction”, “Phenylpropanoid biosynthesis” and “Flavonoid biosynthesis” (Figure 5). On the other hand, submergence treatment results in DEGs enrichment not only in these pathways, but also in “Starch and sucrose metabolism” (Figure 6).

### 2.5. DEGs Screened from Plant Hormone Signal Transduction

Through KEGG enrichment analysis of DEGs, our attention was directed towards the plant hormone signal transduction pathway, which we selected for more in-depth exploration. The results indicated that DEGs were the highest at 5 d under flooding stress. Except for the comparison group Z_15 d vs. 0 d, the number of downregulated genes exceeded that of upregulated genes in all other groups (Table 1). After being subjected to waterlogging stress, 314 DEGs were identified in the roots of *S. japonicus*, while 370 DEGs were detected under submergence. A Venn diagram was constructed to analyze the DEGs involved in plant hormone signal transduction pathway under flooding stress across the five groups (Figure 7). It was found that 54 common DEGs were identified under waterlogging stress (Appendix A), while 112 common DEGs were detected under submergence (Appendix A). Functional annotation indicated that these genes are primarily associated with ethylene, ABA, auxin, and BRs.

### 2.6. DEG Selected from Plant Hormone Biosynthesis Process

#### 2.6.1. Ethylene Biosynthesis

Ethylene biosynthesis is tightly regulated by 1-aminocyclopropane-1-carboxylate synthase (ACS, EC: 4.4.1.14) and ACC oxidase (ACO, EC: 1.14.17.4) [24]. Based on the ethylene biosynthesis GO term (GO:0009693), 16 and 19 genes associated with ethylene biosynthesis were identified from the DEGs under waterlogging stress and submergence, respectively (Table 2). We then also performed a Pearson correlation analysis between the genes on the ethylene biosynthesis process and the content of endogenous hormones (Figure 8). Under flooding stress, *TRINITY_DN20930_c0_g1* (encoding ACS6) was downregulated and correlated negatively with ACC (r = −0.046), while *TRINITY_DN109130_c0_g1* (encoding ACS1) was upregulated. Notably, *TRINITY_DN22884_c0_g1* (encoding ACS1) was uniquely expressed under submergence, with its expression level peaking at 20 d, approximately 3.5 times higher than that at 0 d. Both of the above ACS1 genes were positively correlated with ACC with r values of 0.058 and 0.052. Furthermore, the expression levels of two ACO-encoding genes (*TRINITY_DN4331_c0_g1*, *TRINITY_DN11528_c0_g1*) were significantly higher than those of ACS-encoding genes, exhibiting elevated transcriptional activity that facilitated ethylene biosynthesis. However, the two genes encoding ACO have negative correlation with ACC. In addition, 8 UBA2 genes related to ubiquitylation modification were identified during ethylene biosynthesis process, accounting for nearly half of the total number of genes screened.

#### 2.6.2. ABA Biosynthesis

9-cis-epoxy carotenoid dioxygenase (NCED, EC:1.13.11.51) acts as a key rate-limiting enzyme in the biosynthesis pathway of ABA [25]. Based on the GO term (GO:0009688), we identified 10 genes associated with ABA biosynthesis from the DEGs under waterlogging stress, including 2 genes encoding NCED. Similarly, 10 related genes were also screened from the DEGs under submergence, among which two genes encoding NCED were included (Table 3). Then, a Pearson correlation analysis between the genes on the ABA biosynthesis process and the content of endogenous hormones was performed (Figure 9). Further analysis revealed that the expression of NCED1 (r = 0.4399) was downregulated, NCED2 (r = −0.1331) was specifically expressed only under waterlogging stress, and NCED5 (r = 0.0098) was specifically expressed only under submergence. Within these genes, we also identified three genes encoding ADH1, which is associated with ethanol metabolism in plants under hypoxic conditions.

## 3. Discussion

### 3.1. Ethylene, ABA Biosynthesis and Regulation of Related Genes

Ethylene enhances tolerance by regulating the formation of adventitious roots, the development of aerenchyma, and the induction of plant defense gene expression [26]. ACC is a precursor of ethylene and is influenced by ACS transcript levels [27]. In this study, we found that the expression of ACS gene family members in roots was significantly induced under flooding stress. This has been demonstrated in tomatoes, grapes, cotton and *S. tonkinensis* [28,29,30,31]. In our study, the ACC content initially increased and then decreased, with the decline typically occurring around 10 d, which coincides with the turning point in the expression level of the ACS1 gene. Sequencing of the Arabidopsis genome has revealed that the ACS gene family is encoded by 12 genes, among which ACS1 is predominantly expressed in leaves, while ACS6 is primarily expressed in flowers [32]. In this study, both ACS1 and ACS6 were identified in roots, indicating a shift in the spatiotemporal expression patterns of these genes. This variation may be attributed to factors such as species differences, cis-regulatory elements, transcription factors, and epigenetic modifications. ACC is oxidized to ethylene by ACO [33]. This study identified two highly expressed ACO genes, which may enhance the conversion of ACC to ethylene, thereby accelerating the response of *S. japonicus* to flooding stress. Similarly, *OsACO1-7* were all detected in rice under submerged conditions, and *OsACO1,2,3* and *7* all exhibited moderate-to-high expression levels [34]. Kesawat et al. identified 15 ACO genes, and a variety of cis-acting regulatory elements were identified within the *TaACO* promoters, which are essential for processes related to plant growth, hormone signaling, and responses to defense and stress [35]. In this experiment, the UBA2 gene likely plays a significant role in the ethylene biosynthesis pathway as well. Li et al. examined the promoter regions of 11 *TaUBA2* gene family members and identified 1664 cis-acting elements, among which, the majority were associated with hormone responses, while others were linked to environmental stress, light responses, and developmental processes [36]. However, research on the mechanism of the UBA2 gene in plant responses to flooding stress remains limited, warranting further in-depth investigations.

In addition, ABA, commonly acknowledged as a stress-related phytohormone, shows increased accumulation when plants are exposed to diverse abiotic and biotic stress conditions, and its accumulation leads to stomatal closure to conserve water [37]. However, numerous studies have found that the levels of ABA decrease under flooding stress, which is attributed to ethylene accumulation or the antagonistic interaction with GA [38,39]. The NCED gene cluster plays a pivotal role in regulating abscisic acid production [40]. In this study, we identified NCED genes closely associated with ABA biosynthesis, including NCED1, NCED2, and NCED5. We observed a sharp decline in ABA content in the roots of *S. japonicus* at 5 d under stress conditions. Concurrently, the expression levels of NCED1 decreased by nearly 3-fold and 5-fold under waterlogging stress and submergence, respectively. The results indicate that NCED gene expression is downregulated, leading to a reduction in ABA synthesis. Overexpression of NCED genes increased waterlogging sensitivity in soybeans, which proves, in the opposite case, that ABA depletion facilitates plant acclimation to flooding stress [41]. ADH1 is regulated by ABA-dependent signaling transduction and is strongly induced during flooding in various plant species. In Arabidopsis, exogenous ABA activates ADH1 expression via promoter motifs such as the G-box-1 and GT/GC elements [42]. Papdi et al. further concluded that the stabilized RAP2 transcription factor prolongs ABA-mediated activation of ADH1 [43]. Moreover, the study revealed that ethylene is essential for the induction of ADH during hypoxia, and the expression level of the ADH genes is regulated by local ethylene concentrations [44,45]. This also indicates the crosstalk between hormones.

### 3.2. Interactions Among Ethylene, ABA and GA Under Flooding Stress

Under flooding stress, plants undergo complex hormonal interactions that regulate their growth, development, and survival [46]. The dynamic crosstalk between ABA, ethylene, and GA are essential in regulating how plants adapt to low-oxygen conditions [47].

The findings demonstrate that reduced ABA levels are essential for ethylene and additional stimuli to induce elongation in *Rumex* species; specifically in *R. palustris*, ethylene accumulation promoted growth through the following dual mechanisms: suppression of ABA biosynthesis via downregulation of *RpNCED* expression and acceleration of ABA catabolism into phaseic acid [48]. The transformation of ABA into PA is initiated through the hydroxylation of the 8′ position, facilitated by the enzyme ABA 8′-hydroxylase, which belongs to the cytochrome P450 monooxygenase family [49,50]. Members of the CYP707A gene family are primarily responsible for modulating ABA concentrations in plant systems [50]. In deepwater rice, CYP707A5 was cloned and upregulated significantly by ethylene treatment [51].

It is hypothesized that the increased ethylene concentration leads to a decrease in ABA levels and a concurrent rise in GA content [52]. Ethylene and GA exhibit a cooperative interaction in plants subjected to flooding stress. In deepwater rice, GA enhances both the formation and developmental rate of adventitious roots stimulated by ethylene, dependent on ethylene signaling [53]. Further, it has been found that ethylene promotes internode elongation in response to deep-water environments through activation of the GA synthesis gene SD1 [54]. ABA is considered to be an antagonist of GA [55]. Lin et al. discovered that GA and ABA antagonistically regulate the E3 ubiquitin ligase activity of the APC/CTE complex through SnRK2-APC/CTE, thereby systematically modulating root growth, tillering, and plant height in rice [56]. Likewise, Submergence-induced shoot elongation in *Rumex palustris* is modulated by a reduction in ABA levels and a concurrent rise in GA concentration [57]. In summary, ethylene, ABA, and GA dynamically regulate adaptive responses including aerenchyma formation, and the development of adventitious root, and stem internode elongation through complex antagonistic and synergistic interactions. It is noteworthy that the specific modes of hormone interactions are influenced by factors such as species, stress stages, and intensity.

### 3.3. The Role of Other Hormones in Plants’ Response to Flooding Stress

SA is a small phenolic compound widely present in plants, which participates in various growth and developmental processes such as seed germination, bud emergence, flowering, fruit setting, and fruit ripening [58]. It also plays a crucial role in plant responses to environmental stresses. In the past five years, research on the role of SA in plant responses to abiotic stress tolerance has primarily focused on drought, salinity, heat, cold, and heavy metal stress [59]. For example, in wheat, SA enhances the formation of adventitious roots and aerenchyma while inhibiting root elongation; in tomatoes, exogenous SA application improves shoot growth, leaf number, and chlorophyll content [60,61]. SA is recognized as a key regulator of ROS, and studies have demonstrated that exogenous SA application effectively enhances ROS metabolism and flooding stress tolerance in soybeans by strengthening their antioxidant defense system [62]. There are two pathways of SA biosynthesis in plants, the isochorismate synthase (ICS) pathway and the phenylalanine ammonia-lyase (PYL) pathway [63]. Flooding increases SA content in roots by upregulating specific ICS (*ICS1* and *ICS2*) and PYL (*PYL4*, *PYL5* and *PYL6*) gene expression [60].

Zeatin is a naturally occurring cytokinin in plants, which exists in both cis- and trans-configurations. Among these, trans-zeatin plays a crucial role in regulating plant metabolism. Researchers introduced a SAG12:ipt gene construct (increase cytokinin biosynthesis) into Arabidopsis, and the results showed that SAG12:ipt seedlings rapidly accumulated cytokinins (including trans-zeatin) and exhibited a parallel swift increase in ABA levels, while demonstrating greater biomass, higher carbohydrate content and improved chlorophyll retention [64]. As evidenced by the DEGs enriched in the plant hormone signal transduction pathway in this study, apart from the key hormones measured in this work, auxin, BRs, and jasmonic acid (JA) may also play crucial roles in plant responses to flooding stress.

## 4. Materials and Methods

### 4.1. Materials and Experimental Design

In 2023, the seeds were collected from a 15-year-old mother tree of *Styrax japonicus* planted in the Zhonghua Jasmine Valley, Ma’an Subdistrict, Liuhe District, Nanjing City, Jiangsu Province. The mother tree originated from Rongcheng City, Weihai, Shandong Province. As the final month of the year approached, the seeds were exposed to a low-temperature environment, in sand, with the objective of inducing dormancy. By the end of March 2024, the seeds were sown in trays at the Xiashu Forest Farm of Nanjing Forestry University. After germination, seedlings were transplanted into pots (pot dimensions: 15.6 cm top diameter, 11 cm bottom diameter, and 13.4 cm height) when they reached approximately 15 cm in height. The potting substrate consisted of organic nutrient soil, loess, and perlite in a ratio of 5:3:2. The substrate’s physicochemical characteristics were analyzed, revealing the following measurements: a pH level of 7.14, organic carbon content at 42.7 g/kg, total nitrogen concentration of 2.23 g/kg, total phosphorus at 0.77 g/kg, total potassium amounting to 13.61 g/kg, available nitrogen at 0.18 g/kg, available phosphorus at 21.07 mg/kg, and available potassium at 175.49 mg/kg. In July, the seedlings were moved to the seedling cultivation room at Nanjing Forestry University for acclimatization. In August, consistent growth-performing seedlings were specifically chosen as experimental subjects. Waterlogging stress was defined as water levels 2 cm below the soil surface, while partial submergence was defined as water levels 4 cm above the soil surface (Figure 10). Based on a preliminary experiment conducted in June, it was observed that most *S. japonicus* seedlings remained alive after 25 days under flooding stress. Therefore, the experiment was designed with six sampling time points: 0 d, 5 d, 10 d, 15 d, 20 d, and 25 d. At each time point, 12 seedlings per treatment were destructively sampled, totaling 132 seedlings.

### 4.2. Determination of Endogenous Hormone Content in Roots

The quantification of endogenous plant hormones was performed using electrospray ionization–high performance liquid chromatography–tandem mass spectrometry.

For hormone extraction, weigh out 0.2 g of root samples ground into powder in liquid nitrogen. Then it was determined by using Shimadzu LC-30AD high performance liquid chromatography (Shimadzu, Japan) in series with an AB Company Qtrap6500 mass spectrometer (Wiscosin, USA). The extraction procedures for ACC and the other four hormones differed, as detailed in the table below (Table 4).

Moreover, the detection of ACC differs from that of the other four hormones in terms of liquid chromatography conditions, mass spectrometry parameters, and selected reaction monitoring (SRM) conditions for phytohormones based on protonation or deprotonation modes. The specific parameters are as follows (Table 5):

### 4.3. RNA Extraction and Library Construction

RNA was extracted from the *S. japonicus* roots using TRIzol (Thermo fisher, 15596018). RNA quality was assessed using a NanoDrop ND 1000 (NanoDrop, Wilmington, DE, USA) for concentration and purity, and a Bioanalyzer 2100 (Agilent, CA, USA) for integrity. Samples with concentrations > 50 ng/μL, RIN > 7.0, and total RNA > 1 μg were used for further analysis. mRNA was enriched using Oligo (dT) magnetic beads, fragmented, and converted to single-stranded cDNA with random hexamers. Double-stranded cDNA was synthesized using buffer, dNTPs, RNaseH, and DNA polymerase I, then purified with AMpure XP beads. DNA ends were repaired, A-tailed, and ligated with adapters. USER enzyme degraded the second strand of U-containing cDNA, followed by PCR amplification to construct the sequencing library. The library was quality-checked and sequenced on the Illumina NovaseqTM 6000 (LC Bio-Technology, Hangzhou, China) in PE150 mode.

### 4.4. Quality Control and de Novo Assembly

The sequencing data were cleaned by trimming adapter sequences and discarding low-quality and undetermined bases with Cutadapt (version 1.9) [65]. The quality assessment of the sequence data was performed with FastQC (version 0.10.1), which evaluated key metrics, such as the Q20 and Q30 scores and the GC composition of the processed reads [66]. All subsequent analytical procedures were performed exclusively using rigorously filtered and quality-controlled datasets. The transcriptome was assembled de novo using the Trinity software (version 2.15) [67].

### 4.5. KEGG Enrichment Analysis

The differentially expressed Unigenes (DEGs) were identified using R package edgeR (3.40.2) with a threshold set at log2 (fold change) >1 or <−1 and a false discovery rate (FDR) < 0.05. All significant differential expression genes were mapped toward the KEGG annotation results for each pathway, the number of Unigenes per pathway was calculated, and then the hypergeometric test was applied to identify KEGG pathways that were significantly enriched in significant DEGs compared to the KEGG annotation results of the entire Unigenes.

### 4.6. qRT-PCR Analysis

A total of 8 DEGs related to plant hormones were selected to be validated through qRT-PCR. The primer sequences are shown in Appendix A, primers used for quantitative real-time PCR (qRT-PCR) analysis (Appendix A). Based on previous studies, GADPH served as the internal control for normalization. StepOne Real-Time PCR System (Applied Biosystems, Foster City, CA, USA) and SYBR Green Premix Pro Taq HS qPCR Kits (AG11701, Accurate Biotechnology, Hunan, Co., Ltd., Changsha, Hunan, China) were used [68]. The quantification of gene expression was determined by employing the 2^−∆∆Ct^ method (Appendix A).

### 4.7. Statistical Analysis

Data on endogenous plant hormones were analyzed by chi-square test and ANOVA one-way analysis of variance using SPSS 25 (IBM Corp., Armonk, NY, USA). The homogeneity of variances was tested using Levene’s test. If the assumption of equal variance was met (*p* > 0.05), Tukey’s HSD post hoc test was conducted to determine pairwise differences. If the assumption was violated (*p* < 0.05), Tamhane’s T2 test was used instead. Statistical significance was considered at *p* < 0.05. All data are expressed as means ± standard deviation (SD), and significant differences among groups were indicated using different lowercase letters above the error bars. Graphs were plotted using Origin 2024b (OriginLab Corporation, Northampton, MA, USA).

## 5. Conclusions

This research explored the adaptive mechanisms of *S. japonicus* under flooding stress by measuring the endogenous hormone content in its roots and conducting transcriptome sequencing analysis. The results demonstrated that *S. japonicus* roots adapt to flooding stress by regulating the synthesis and signal transduction of plant hormones such as ethylene, ABA, and GA. The trend in ACC content indicated that *S. japonicus* initiates the ethylene response mechanism earlier under submergence, while the dynamic changes in ABA content reflected greater ABA consumption under submergence compared to waterlogging stress. A significant decrease in GA content suggested that flooding stress inhibits plant growth activity. Transcriptome analysis identified numerous genes associated with the signaling pathways and metabolic processes of ethylene and ABA, among which the genes encoding ACS, ACO, and NCED enzymes were particularly critical. Further research, including gene functional validation and molecular interaction mechanism analysis, can be conducted based on these findings. Our findings offer fundamental insights for elucidating the regulatory network of *S. japonicus* in response to waterlogging stress and for breeding waterlogging-tolerant varieties.

## Figures and Tables

**Figure 1 plants-14-01870-f001:**
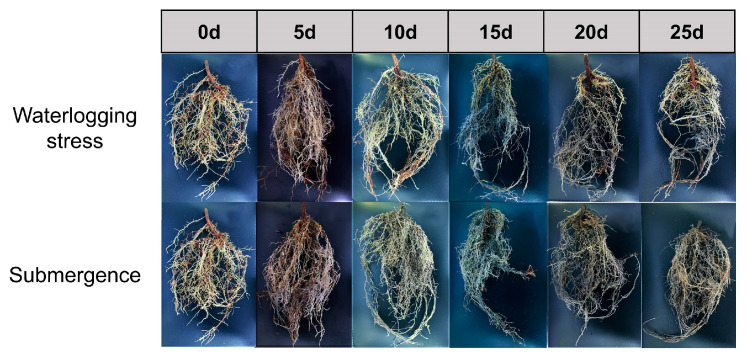
Morphological changes in the root system of *S. japonicus* at various sampling time points.

**Figure 2 plants-14-01870-f002:**
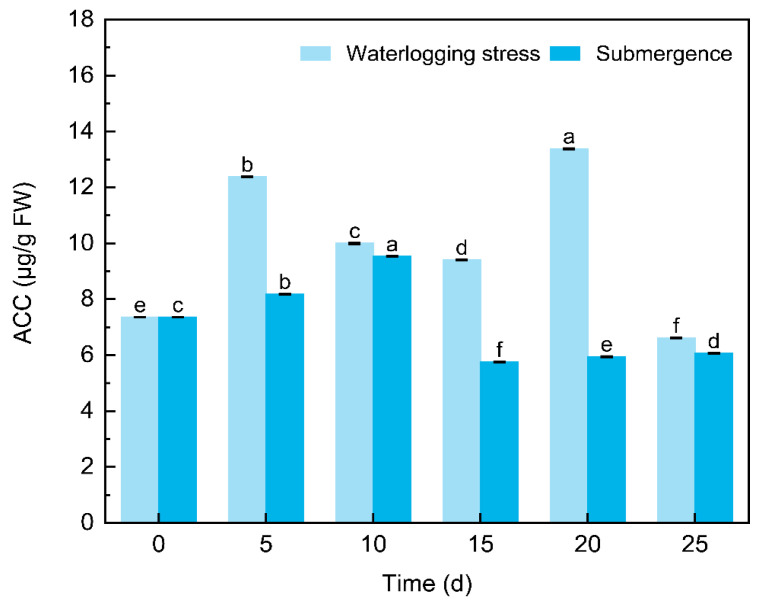
Changes in ACC content in the roots of *S. japonicus* under flooding stress**.** Lowercase letters indicate the significance of differences between different sampling times under the same treatment. The same below.

**Figure 3 plants-14-01870-f003:**
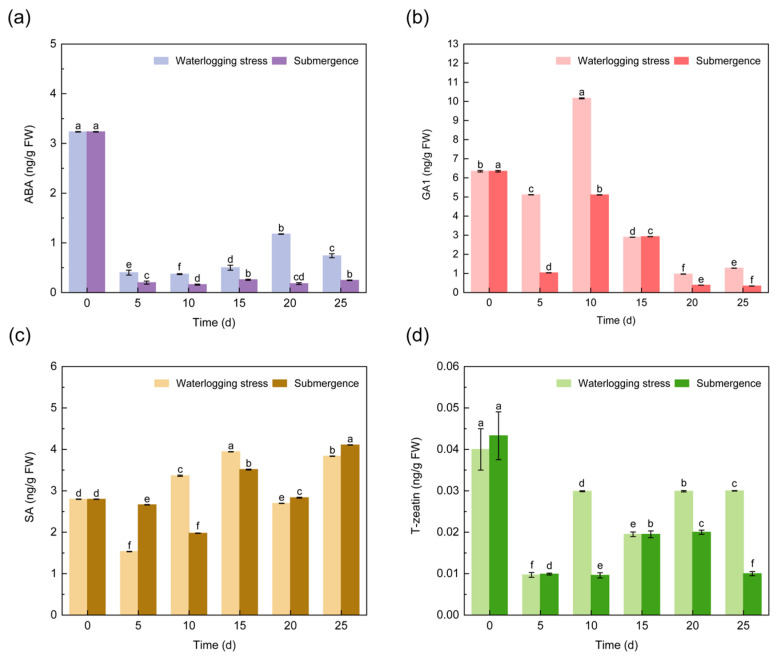
Changes in ABA (**a**), GA1 (**b**), SA (**c**), and T-Zeatin (**d**) content in the roots of *S. japonicus* under flooding stress.

**Figure 4 plants-14-01870-f004:**
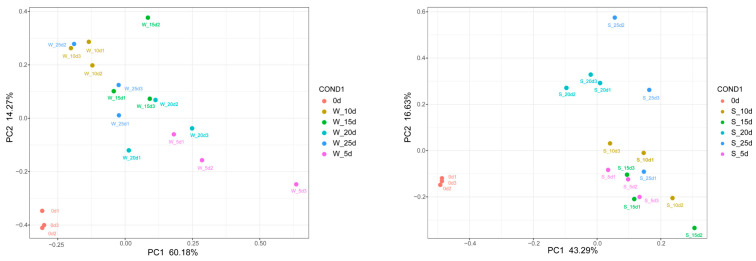
PCA analysis of samples under flooding stress. “W” and “S” represent waterlogging stress and submergence, respectively.

**Figure 5 plants-14-01870-f005:**
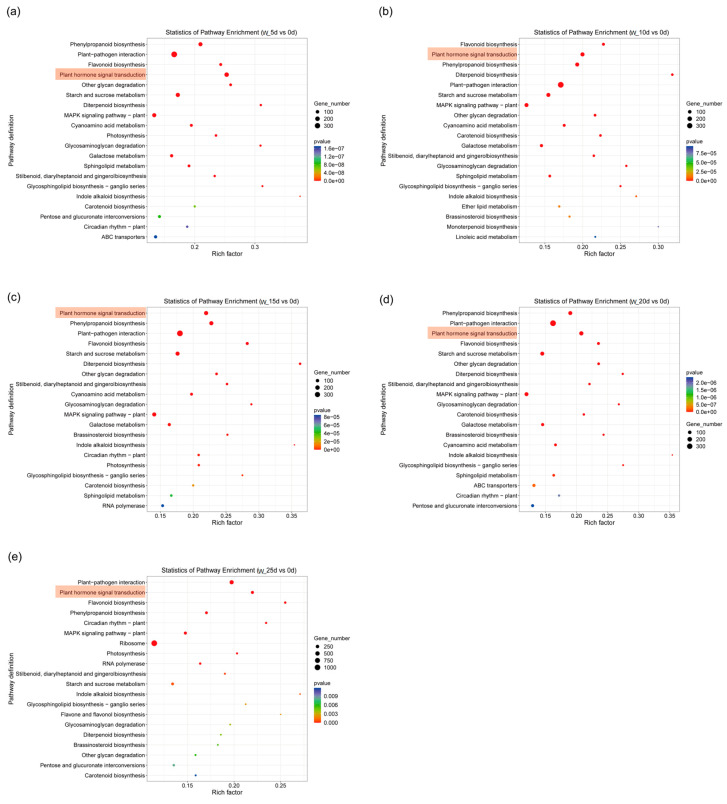
KEGG pathway analysis of DEGs; comparison between 5 d (**a**), 10 d (**b**), 15 d (**c**), 20 d (**d**), 25 d (**e**) and 0 d under waterlogging stress.

**Figure 6 plants-14-01870-f006:**
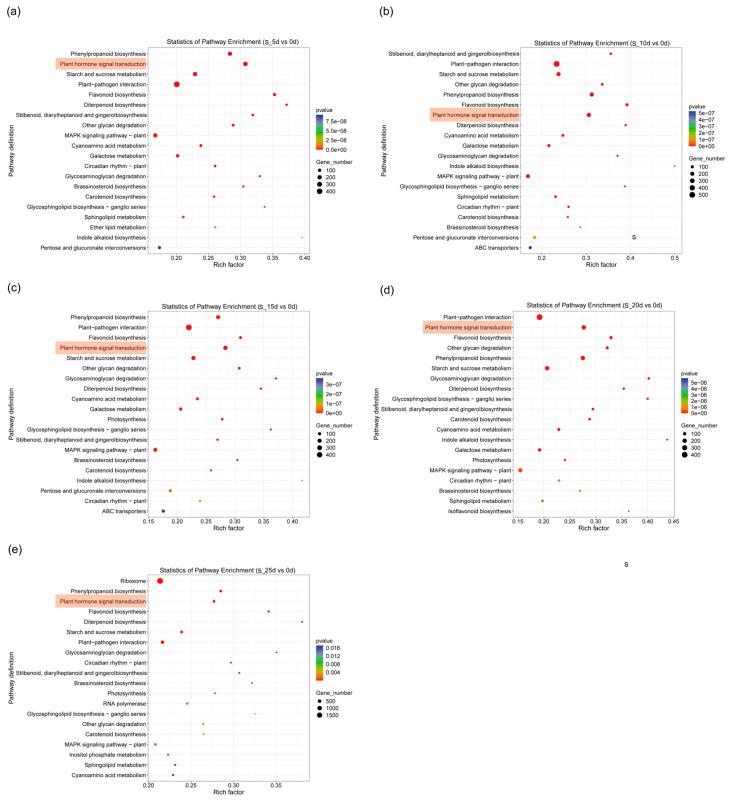
KEGG pathway analysis of DEGs; comparison between 5 d (**a**), 10 d (**b**), 15 d (**c**), 20 d (**d**), 25 d (**e**) and 0 d under submergence.

**Figure 7 plants-14-01870-f007:**
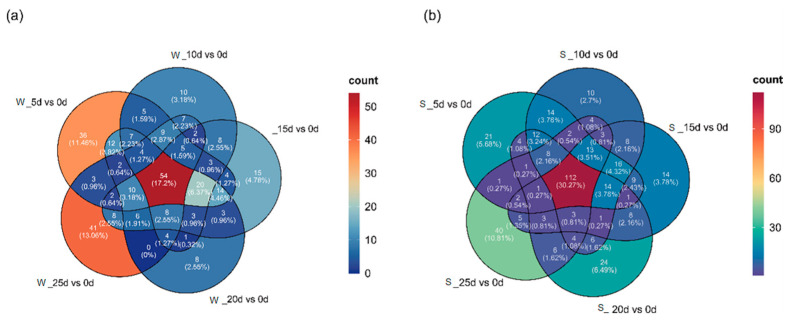
Venn plots of DEGs on the plant hormone signal transduction pathway under waterlogging stress (**a**) and submergence (**b**).

**Figure 8 plants-14-01870-f008:**
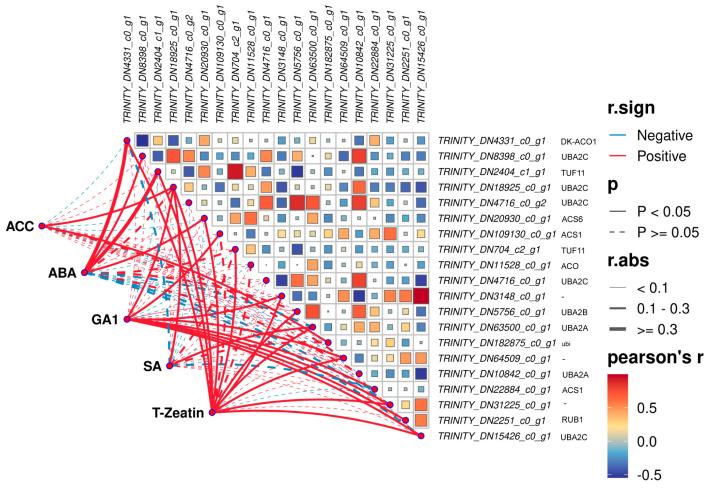
Correlation analysis of DEGs on the ethylene biosynthesis process and endogenous hormone content under flooding stress.

**Figure 9 plants-14-01870-f009:**
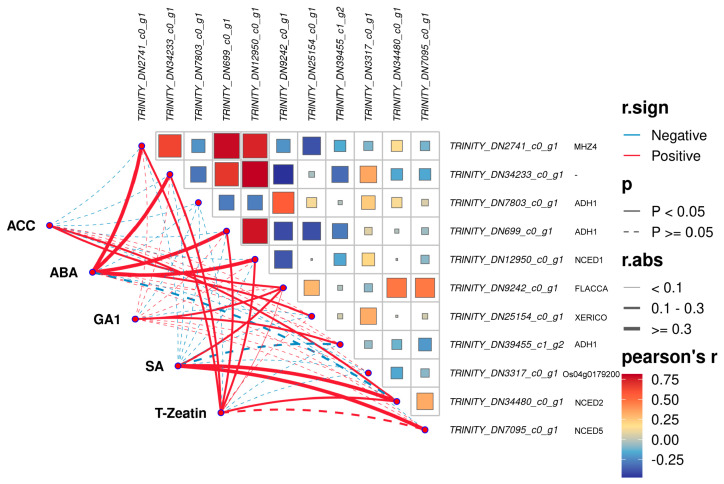
Correlation analysis of DEGs on the ABA biosynthesis process and endogenous hormone content under flooding stress.

**Figure 10 plants-14-01870-f010:**
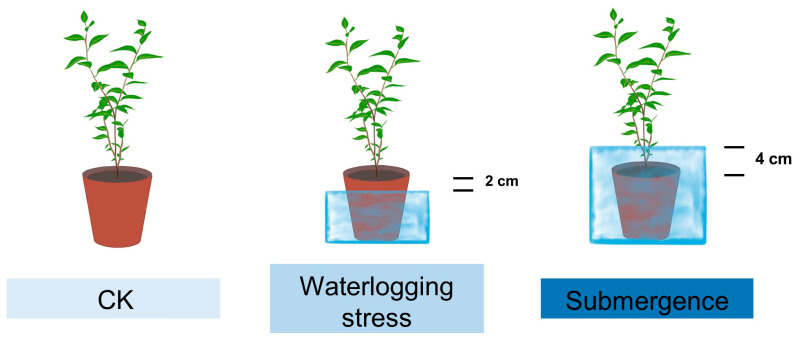
Schematic diagram of flooding stress treatment.

**Table 1 plants-14-01870-t001:** Total number of DEGs and number of up- and downregulated genes under flooding stress.

Group	Total	Up	Down
Z_5 d vs. 0 d	190	72	118
Z_10 d vs. 0 d	150	66	84
Z_15 d vs. 0 d	165	83	82
Z_20 d vs. 0 d	156	68	88
Z_25 d vs. 0 d	165	78	87
L_5 d vs. 0 d	231	94	137
L_10 d vs. 0 d	230	84	146
L_15 d vs. 0 d	213	75	138
L_20 d vs. 0 d	208	88	120
L_25 d vs. 0 d	208	82	126

**Table 2 plants-14-01870-t002:** DEGs involved in the ethylene biosynthesis pathway under flooding stress.

	Gene Id	Annotation	Name
Common DEGs	TRINITY_DN4331_c0_g1	ACC oxidase 5 [*Actinidia deliciosa*]	DK-ACO1
	TRINITY_DN8398_c0_g1	hypothetical protein HYC85_014601 [*Camellia sinensis*]	UBA2C
	TRINITY_DN2404_c1_g1	ubiquitin-40S ribosomal protein S27a isoform X3 [*Elaeis guineensis*]	TUF11
	TRINITY_DN18925_c0_g1	UBP1-associated protein like [*Actinidia chinensis* var. chinensis]	UBA2C
	TRINITY_DN4716_c0_g2	UBP1-associated protein 2C-like [*Diospyros lotus*]	UBA2C
	TRINITY_DN20930_c0_g1	hypothetical protein RHSIM_Rhsim07G0098100 [*Rhododendron simsii*]	ACS6
	TRINITY_DN109130_c0_g1	hypothetical protein HYC85_011085 [*Camellia sinensis*]	ACS1
	TRINITY_DN704_c2_g1	ubiquitin-40S ribosomal protein S27a isoform X3 [*Elaeis guineensis*]	TUF11
	TRINITY_DN11528_c0_g1	1-aminocyclopropane-1-carboxylate oxidase-like [*Actinidia eriantha*]	ACO
	TRINITY_DN4716_c0_g1	UBP1-associated protein 2C [*Camellia lanceoleosa*]	UBA2C
	TRINITY_DN3148_c0_g1	bi-ubiquitin [*Scenedesmus* sp. NREL 46B-D3]	-
	TRINITY_DN5756_c0_g1	UBP1-associated protein 2B [*Camellia lanceoleosa*]	UBA2B
	TRINITY_DN63500_c0_g1	hypothetical protein F0562_016212 [*Nyssa sinensis*]	UBA2A
	TRINITY_DN182875_c0_g1	UBQ10 [*Scenedesmus* sp. PABB004]	ubi
	TRINITY_DN64509_c0_g1	polyubiquitin-like [*Hordeum vulgare* subsp. vulgare]	-
Unique to waterlogging stress	TRINITY_DN10842_c0_g1	UBP1-associated protein 2A-like [*Camellia sinensis*]	UBA2A
Unique to submergence	TRINITY_DN22884_c0_g1	hypothetical protein C3L33_16073, partial [*Rhododendron williamsianum*]	ACS1
	TRINITY_DN31225_c0_g1	hypothetical protein COHA_002734 [*Chlorella ohadii*]	-
	TRINITY_DN2251_c0_g1	LOW QUALITY PROTEIN: polyubiquitin-A-like [*Panicum hallii*]	RUB1
	TRINITY_DN15426_c0_g1	hypothetical protein CBR_g19053 [*Chara braunii*]	UBA2C

**Table 3 plants-14-01870-t003:** DEGs involved in the ABA biosynthesis pathway under flooding stress.

	Gene Id	Annotation	Name
Common DEGs	TRINITY_DN2741_c0_g1	hypothetical protein HYC85_002681 [*Camellia sinensis*]	MHZ4
	TRINITY_DN34233_c0_g1	hypothetical protein LOK49_LG10G01334 [*Camellia lanceoleosa*]	-
	TRINITY_DN7803_c0_g1	hypothetical protein HHK36_032021 [*Tetracentron sinense*]	ADH1
	TRINITY_DN699_c0_g1	hypothetical protein RHGRI_006931 [*Rhododendron griersonianum*]	ADH1
	TRINITY_DN12950_c0_g1	hypothetical protein F0562_010481 [*Nyssa sinensis*]	NCED1
	TRINITY_DN9242_c0_g1	Molybdenum cofactor sulfurase [*Camellia lanceoleosa*]	FLACCA
	TRINITY_DN25154_c0_g1	hypothetical protein RHMOL_Rhmol04G0049500 [*Rhododendron molle*]	XERICO
	TRINITY_DN39455_c1_g2	secoisolariciresinol dehydrogenase-like [*Camellia sinensis*]	ADH1
	TRINITY_DN3317_c0_g1	momilactone A synthase-like [*Camellia sinensis*]	Os04g0179200
Unique to waterlogging stress	TRINITY_DN34480_c0_g1	hypothetical protein LOK49_LG07G02934 [*Camellia lanceoleosa*]	NCED2
Unique to submergence	TRINITY_DN7095_c0_g1	carotenoid 9,10(9′,10′)-cleavage dioxygenase 1-like [*Camellia sinensis*]	NCED5

**Table 4 plants-14-01870-t004:** The hormone extraction steps.

ACC	ABA, GA1, SA, T-Zeatin
1. Add ddH_2_O, pre-cooled to 4 °C, and extract at 4 °C for 2 h; 2. Centrifuge 10,000× *g* at 4 °C for 5 min, take the supernatant, extract the precipitate once again, combine the supernatants, pass through MCX pre-loading, and elute with 8 mL of water; 3. Pass through a 0.22 µm filter membrane and place in a 4 °C refrigerator for on-machine testing.	1. Extract at 4 °C overnight, centrifuge at 12,000× *g* for 5 min, and take the supernatant; 2. Add five times the volume of acetonitrile solution to the precipitate again, extract twice, and combine to obtain the supernatant; 3. Pass through a C18 solid–phase extraction column, shake vigorously for 30 s, centrifuge at 10,000× *g* for 5 min, and take the supernatant; 4. Vacuum centrifugal concentration until dry, resolubilize with 200 μL methanol, pass through a 0.22 μm filter membrane, and place in a −20 °C refrigerator for on-machine testing.

**Table 5 plants-14-01870-t005:** HPLC-MS/MS protocol for hormone quantification.

Condition	Name	ACC	ABA, GA1, SA, T-Zeatin
liquid chromatography conditions	Chromatographic column	Poroshell 120 SB-C18 Reversed phase column (2.1 × 150, 2.7 um)	
	Column temperature	35 °C	30 °C
	Mobile phase	A:B = acetonitrile: (water/0.1% formic acid) = 3:7	A:B = (methanol/0.1% formic acid): (water/0.1% formic acid)
	Elution gradient	Isometric gradient elution	Appendix A
	Injection volume	2 µL	
mass spectrometry parameters	Ionization mode	ESI positive ion mode	ESI positive and negative ion modes are monitored separately
	Scanning type	MRM	
	curtain gas	15 psi	
	Spray voltage	+4500 v	+4500 v, −4500 v
	Atomized gas pressure	65 psi	
	Auxiliary gas pressure	70 psi	
	Atomization temperature	350 °C	400 °C

## Data Availability

The transcriptome data used in this article is uploading now. It will be added as soon as it is successfully uploaded.

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
