# Peer review of "Root Ethylene and Abscisic Acid Responses to Flooding Stress in Styrax japonicus: A Transcriptomic Perspective"

_plants, 2025, doi:10.3390/plants14121870_

Round 1
Reviewer 1 Report
Comments and Suggestions for Authors
The topic of the MS is interesting and novel, focusing on the metabolic, especailly the hormonal changes and alteration in the transcriptome in Styrax japonicus after flooding stress.
The abstract and introduction are clear and detailed enough, and due to the logical summary of the literature, the main aim is understandable. Just one suggestion, in line mentione more interaction between plant hormones, e.g. SA and ABA general antagonism, etc.
Please try to aviod references available only in Chinese.
M and M section: please complete the growth conditions with the information about the plants, not only the sowing and treatment date, but the age of plants e.g. 60-day-old plants. When the control samples were collected, only on the 0d? As a 25-day long experimental duration can alone influence the hormon content, they have even a cyrdian rhythm, so be careful. If this is true, the results should be rather compared to each other (waterlogging to submergence) than to the absolute control in the text. And try to avoid statement like this “increase-decrease-increase” as no adequate controls.
Highlight more that only the roots were analysed for plant hormones, and transcriptome.
Figure 7 and 8 the combination of the results from all the treatments? I mean waterlogging and submergence?
Please correct "ethylene" in the beging of some sentences to "Ethylene".
In 264 and 283 "Rumex" in Italics.
Discussion and conclusion is quite short, but supported by the results. Please complete it with some further aspects.
Reviewer 2 Report
Comments and Suggestions for Authors
The manuscript “Root Hormone Responses to Flooding Stress in Styrax japonicus: A Transcriptomic Perspective” is focused on elucidation of the role of ethylene, ABA, GA, SA and t-Zeatin in plant roots adaptation to oxygen deprivation. Presented results could be important in theoretical and practical sense because of the multifunctional usage of S. japonicas, including its valuability for medicine. Authors have been used several well established methods to determine phytohormone concentration and gene expression intensity. All together the manuscript subject, used methods and obtained data fit the journal scope.
Nevertheless number of questions is rising, some of them are crucial:
- Material and Method section has to be improved. One of the very important questions is how root samples were collected? Which part of the root was analyzed? How intensive was damage in root development during experiment?
- Another series of question concern section describing plant hormones detection. All the tested hormones have different procedure of analysis, thus appropriate information has to be added and proper citation of methods used is needed.
- Discussion section mostly focused on the role of ethylene and ABA. The level of SA and t-Zeatin also affected by oxygen lack but these results are not discussed. Why authors choose GA1 from among several active substances of this group? According to KEGG pathway analysis one of the DEGs group was “Plant hormone signal transduction” but it is not discussed how it corresponds to hormone synthesis and/or mechanism of its action during oxygen limitation. Were these genes linked only with ethylene and ABA? Some additional information is needed for the description of UBA2 group with proper citation. It is also under the question the involvement of ADH1 protein in ABA synthesis. In addition the level of ACC and GA1 had a non-linear dynamic and was quite different at waterlogging and submergence. It would be of importance to hypothesize the reason and outcome of that and how it could correspond to changes in growth of japonicas roots.
Some remarks can be addressed to punctuation marks, the absence of capital letters at the beginning of phrases and headings, to several citations in the list.
Taken together I can conclude that in present form manuscript needs major revision.
Reviewer 3 Report
Comments and Suggestions for Authors
MS titled "Root hormone responses to flooding stress in Styrax japonicus: A transcriptomic perspective" presents fluctuations of several hormones and transcriptome under flooding stress. I have several main points: 1. I miss data on root biomass and morphology under stress conditions. 2. Auxins are very important for root development, but authors did not measure art least IAA. 3. There are some changes in auxin responsive genes but it was not elaborated. 4. Analytical methodology is very poor, and need description and reference. 5. According my opinion each experimental point needs to have own control. 0 day control is not appropriate for 25 days of treatment. 6. Some data from suppl. mat. are more appropriate to include in article, and some long tables in suppl. mat. 7. It is more appropriate to use gene name than gene ID in figures. It is more reader friendly.
Considering results, I do not see scientific novelty, or authors did not present it on adequate way. Thus, I can not recommend this MS for publication.

MS need some improvement.
Round 2
Reviewer 2 Report
Comments and Suggestions for Authors
Authors represent new version of the manuscript “Root Hormone Responses to Flooding Stress in Styrax japonicus: A Transcriptomic Perspective”, which was revised according to reviewer’s remarks.
At present form manuscript can be accepted.
Author Response
Dear reviewer,
Thank you for your professional advice. It's a great honor for me to have your support. I will continue to make serious improvements based on the editor's suggestions and the format required for publication. Thank you again.